# Numerical Simulation of Micro-Bubbles Dispersion by Surface Waves

**Oleg A. Druzhinin [1],\* and Wu-Ting Tsai [2]**

1    Institute of Applied Physics of the Russian Academy of Sciences, 603950 Nizhny Novgorod, Russia
2    Department of Engineering Science and Ocean Engineering, Taiwan National University, Taipei 106, Taiwan; wttsai@ntu.edu.tw
\*    Correspondence: druzhinin@ipfran.ru

**Abstract:** This paper presents an algorithm for numerical modeling of bubble dispersion occurring in the near-surface layer of the upper ocean under the action of non-breaking two-dimensional (2D) surface waves. The algorithm is based on a Eulerian-Lagrangian approach where full, 3D Navier-Stokes equations for the carrier flow induced by a waved water surface are solved in a Eulerian frame, and the trajectories of individual bubbles are simultaneously tracked in a Lagrangian frame, taking into account the impact of the bubbles on the carrier flow. The bubbles diameters are considered in the range from 200 to 400 microns (thus, micro-bubbles), and the effects related to the bubbles deformation and dissolution in water are neglected. The algorithm allows evaluation of the instantaneous as well as statistically stationary, phase-averaged profiles of the carrier-flow turbulence, bubble concentration (void fraction) and void-fraction fluxes for different flow regimes, both with and without wind-induced surface drift. The simulations results show that bubbles are capable of enhancing the carrier-flow turbulence, as compared to the bubble-free flow, and that the vertical water velocity fluctuations are mostly augmented, and increasingly so by larger bubbles. The results also show that the bubbles dynamics are governed by buoyancy, the surrounding fluid acceleration force and the drag force whereas the impact of the lift force remains negligible.

**Keywords:** numerical simulation; surface water waves; bubble dispersion; turbulence modification



## 1. Introduction

Understanding the dynamics of small-scale processes occurring in the vicinity of the sea surface is important for modeling the exchange of mass, heat and momentum between the atmosphere and the ocean [1,2]. In many practical situations, clouds of bubbles, produced mainly by breaking surface waves, may also affect the state of the near-surface water layer [3,4]. Laboratory and field observations [3,5–7] as well as recent direct numerical simulations of breaking waves [8] indicate that whereas comparatively large bubbles (with diameters $d > 1$ mm) quickly rise to the surface and burst, smaller (or micro-) bubbles ($d \sim 100$ μm) are suspended in water for a considerably longer times and thus contribute mostly to the void fraction observed in the near-surface oceanic layer.

According to field observations the concentration of microbubbles (and thus the void fraction) in the upper ocean layer can be considerable even at relatively low winds and affects gas exchange between the air and water [9,10], production of sea-salt aerosols [11], and the propagation of sound in the upper ocean [12,13]. Therefore, modeling microbubble dynamics is important for many practical applications.

Numerical modeling of the dispersion of microbubbles by surface waves and currents taking into account their impact on the near-surface turbulence represents a challenging problem, and various models are employed in numerical investigations of bubbly flows [14]. Known numerical studies of microbubble dispersion are mainly restricted to either bubble-laden isotropic turbulence (with periodic boundary conditions) [15], or boundary-layer

flows in the vicinity of a fixed, flat boundary [16]. However, performing numerical simulation of a flow in the vicinity of a waved boundary involves additional efforts required to resolve a strong geometric nonlinearity. There are mainly two methods employed to cope with this problem in numerical studies. One approach is related to using the Volume of Fluid method where the air and water phases are directly resolved (cf., e.g., [8]). However, in order to model a sufficiently high void fraction (on the order of $10^{-5}$ [12]), the number of microbubbles to be considered in the framework of a fully-resolved numerical experiment with an adaptive mesh refinement may become prohibitively large. Another approach (also used and discussed in the present study) employs a mapping of the physical coordinates onto the curvilinear coordinates where the waved surface is reduced to a flat surface.

The objective of the present paper is to present a numerical algorithm for evaluation of the dispersion of micro-bubbles by progressive, non-breaking surface (Stokes) waves. The wave shape is prescribed and assumed to be stationary, and unaffected by either bubbles or induced turbulent motions. Thus it is assumed that for typical void fractions observed in the upper ocean layer (on the order of $O(10^{-5})$ or less), the impact of bubbles on the carrier surface wave flow remains negligible. Results of numerical experiments also reveal that typical amplitudes of turbulent wave-induced fluctuations are by orders of magnitude smaller as compared to the energy-containing mother wave [17]. However, in the present study, the impact of bubbles on the induced turbulence, although not so significant at the considered void fractions, is accounted for. The algorithm also allows to investigate how wind-induced surface drift affects bubbles dynamics employing a parameterization of the drift velocity at the water surface in terms of the air friction velocity (determined by the velocity scale tied to the surface wave celerity) [18]. The numerical method is based on the algorithms previously developed for modeling atmospheric boundary layer over progressive surface waves under various conditions (stable air-stratification, parasitic capillaries, droplets) [19–23] and adapted here for modeling a bubble-laden near-surface water layer.

## 2. Governing Equations

Figure 1 outlines the schematic of the problem under consideration.

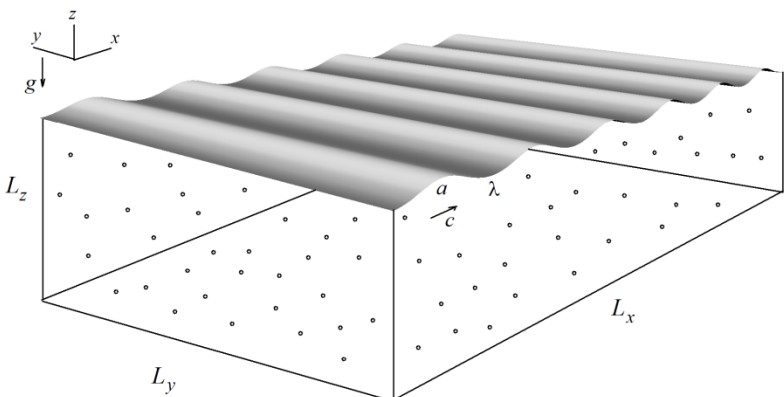

**Figure 1.** Schematic of the problem: $(x, y, z)$ are the Cartesian coordinates; $a$, $\lambda$, $c$ are the surface wave amplitude, length, and celerity; $g$ is the acceleration due to gravity; $L_x$, $L_y$, $L_z$ are the domain sizes. Symbols (open circles) show injected bubbles (not to scale).

A domain with sizes $L_x = 6\lambda$, $L_y = 4\lambda$, $L_z = \lambda$, with periodic side boundaries, a solid bottom and a waved upper boundary, $z_s(x, t)$, is considered, and the Cartesian coordinates are employed, $0 \leq x \leq L_x$; $-L_y/2 \leq y \leq L_y/2$; $-L_z \leq z \leq z_s$. The motion of the fluid (water) is driven by the upper boundary where a progressive, two-dimensional (2D) stationary wave of amplitude $a$, celerity $c$ and length $\lambda$ propagating in the positive $x$-direction is prescribed.

A Eulerian-Lagrangian approach is adopted where the Navier-Stokes equations of the water motion are solved in a Eulerian frame, and the bubbles are tracked simultaneously by solving their respective equations of motion in a Lagrangian frame.

The Navier-Stokes equations for the carrier fluid are written in the dimensionless form [24]:

$$\frac{\partial U_i}{\partial t} + \frac{\partial (U_i U_j)}{\partial x_j} = -\frac{\partial P}{\partial x_j} + \frac{1}{\text{Re}} \frac{\partial^2 U_i}{\partial x_j \partial x_j} + \sum_{n=1}^{N_b} S_i^n, \tag{1}$$

where $x_i \equiv (x, y, z)$, $U_j = (U_x, U_y, U_z)$ are water velocity components, $P$ is the pressure, and $S_i^n$ is a momentum source term (cf. Equation (8) below) contributed by the $n$-th bubble, and $n = 1, \dots, N_b$, the latter being the total, constant, number of tracked bubbles. Variables in Equation (1) are normalized with velocity scale, $U_0$, set equal the surface wave phase speed, $c$, and length scale, $L_0$, equal to the wave length, $\lambda$. The pressure is normalized with $\rho U_0^2$ where $\rho$ is the water density ($\approx 1$ g/cm$^3$). The carrier flow Reynolds number, Re, measures the ratio of the inertial vs. viscous effects and is defined as:

$$\text{Re} = \frac{U_0 L_0}{\nu}, \tag{2}$$

where $\nu$ is the water kinematic viscosity ($\approx 0.01$ cm$^2$/s). Equation (1) is supplemented by the incompressibility condition:

$$\frac{\partial U_j}{\partial x_j} = 0, \tag{3}$$

which implicitly defines the pressure field, $P$ (cf. Equation (18) below).

The bubbles equations of motion are written as [15]:

$$\frac{dr_i^n}{dt} = V_i^n, \tag{4}$$

$$\frac{dV_i^n}{dt} = 3\frac{DU_i^n}{Dt} + \frac{f(\text{Re}_b^n)}{\tau_n}(U_i^n - V_i^n) + \frac{1}{2}\,\varepsilon_{ijk}\left(U_j^n - V_j^n\right)\omega_k^n + 2g\delta_{iz}. \tag{5}$$

In Equations (4) and (5), $r_i^n$, $V_i^n (i = x, y, z)$ are the $n$-th bubble coordinate and velocity components, $U_i^n$ is the fluid velocity at the bubble location, $d/dt = \partial/\partial t + V_j^n \partial_j$ and $D/Dt = \partial/\partial t + U_j^n \partial_j$ are the material derivatives along the bubble trajectories and the surrounding fluid Lagrangian paths, respectively; $g$ is the dimensionless gravitational acceleration, and $\omega_i^n = \varepsilon_{ijk}\partial_j U_k^n$ is the surrounding fluid vorticity; $\delta_{ij}$ and $\varepsilon_{ijk}$ are the Kronecker and Levi-Civita tensors. The forces acting on the bubble (in the order as in the right hand side of Equation (5)) are the fluid acceleration, viscous drag, lift, and buoyancy. The correction in the viscous drag force (accounted for by factor $f$) is caused by a finite Reynolds number of the bubble and defined as:

$$\text{Re}_b^n = \frac{d|U^n - V^n|}{\nu}, \tag{6}$$

where [15],

$$f(\text{Re}_b) = 1 + 0.197\text{Re}_b^{0.63} + 2.6 \times 10^{-4}\text{Re}_b^{1.38}. \tag{7}$$

Both the momentum source term in the right hand side of Equation (1), $S_i^n$, and Equation (5) are formulated under an assumption of a negligible mass of the bubble as compared to water. A "point force" approximation is adopted, whereby the following formulation for the source term, $S_i^n$, is employed [23]:

$$S_i^n = \frac{\pi d_n^3}{6}\left(\frac{DU_i^n}{Dt} + g\delta_{iz}\right)\frac{w(r^n, r)}{\Omega_g}, \tag{8}$$

where $w(r^n, r)$ is a geometrical weight-factor inversely proportional to the distance be-tweeen the *n*-th bubble located at $r^n = (x^n, y^n, z^n)$ and the grid node at $r = (x, y, z)$, and $\Omega_g$ $(r)$ is the volume of the considered grid cell. Thus, for each individual bubble, eight weight-factors are defined (for each of the surrounding grid nodes) and normalized, so that the sum of partial contributions distributed to these nodes exactly equals the respective total source contribution. Therefore, there is no numerically induced loss or gain of momentum in the bubble-water exchange processes (cf., e.g., [23] and references therein for a more detailed discussion).

## 3. Numerical Method

In order to avoid coping with a strong geometric nonlinearity caused by the wavy upper boundary during the integration of the governing equations discussed above, a mapping is introduced transforming the domain with a wavy upper boundary into a do-main with a flat upper boundary, and relating the Cartesian coordinates $(x, z)$ to curvilinear coordinates $(\xi, \eta)$ as:

$$
\begin{aligned}
\xi &= x + a \exp(kz) \sin k(x - ct) \\
\eta &= z + a \exp(kz) \cos k(x - ct)
\end{aligned}
\tag{9}
$$

Mapping (9) transforms the wavy (upper) boundary at $z = z_s(x, t)$ into a flat-plane boundary at $\eta = 0$. The water surface elevation, $z_s(x, t)$, is defined implicitly by Equation (9) and up to the second order in $ka$ coincides with the Stokes-wave solution [25]:

$$
z_s(x, t) \approx -a \cos k(x - ct) + \frac{a^2 k}{2} \cos 2k(x - ct).
\tag{10}
$$

An additional mapping is also employed for the vertical coordinate in the form:

$$
\widetilde{\eta} = \tanh \eta,
\tag{11}
$$

where $-1 \le \widetilde{\eta} \le 0$, so that the grid nodes are clustered in the vicinity of the upper boundary, at $\widetilde{\eta} = \eta = 0$, and stretched with increasing depth.

Since the mapping, Equation (9), is conformal, the following relations between the derivatives hold:

$$
\frac{\partial \xi}{\partial x} = J \frac{\partial x}{\partial \xi} = J \frac{\partial z}{\partial \eta} = \frac{\partial \eta}{\partial z}; \quad \frac{\partial \xi}{\partial z} = -J \frac{\partial x}{\partial \eta} = J \frac{\partial z}{\partial \xi} = -\frac{\partial \eta}{\partial x},
\tag{12}
$$

where the Jacobian of the transformation is:

$$
J = \left( \frac{\partial \xi}{\partial x} \right)^2 + \left( \frac{\partial \xi}{\partial z} \right)^2.
\tag{13}
$$

Due to the properties in Equations (12) and (13), the derivatives over the Cartesian coordinates, *x* and *z*, can be related to the derivatives over curvilinear coordinates, $\xi, \eta$, as,

$$
\begin{aligned}
\frac{\partial}{\partial x} &= \frac{\partial \xi}{\partial x} \frac{\partial}{\partial \xi} + \frac{\partial \eta}{\partial x} \frac{\partial}{\partial \eta} = J \left( \frac{\partial x}{\partial \xi} \frac{\partial}{\partial \xi} + \frac{\partial x}{\partial \eta} \frac{\partial}{\partial \eta} \right), \\
\frac{\partial}{\partial z} &= \frac{\partial \xi}{\partial z} \frac{\partial}{\partial \xi} + \frac{\partial \eta}{\partial z} \frac{\partial}{\partial \eta} = J \left( \frac{\partial z}{\partial \xi} \frac{\partial}{\partial \xi} + \frac{\partial z}{\partial \eta} \frac{\partial}{\partial \eta} \right).
\end{aligned}
\tag{14}
$$

The Laplacian operator is also rewritten as:

$$
\frac{\partial^2}{\partial x_j \partial x_j} = J \left( \frac{\partial^2}{\partial \xi^2} + \frac{\partial^2}{\partial \eta^2} \right) + \frac{\partial^2}{\partial y^2}
\tag{15}
$$

Equations (1) and (3) for the water velocity are discretized on a staggered grid con-sisting of $360 \times 240 \times 180$ nodes, in the $\xi$, *y* and $\widetilde{\eta}$ coordinate directions using a second-order-accuracy, finite-difference method. At each time moment, $t_k$, the mesh is redefined and adapted to the shape of the water surface, $z_s(x, t_k)$, according to Equation (9), and all

fields computed during the preceding time step are re-defined on the new mesh by a linear interpolation.

The integration of Equation (1) is advanced in time by the second-order-accuracy Adams-Bashforth method in two stages to calculate the water velocity at each new time step, $U_i(t_{k+1})$. First, an intermediate velocity, $U_i^*$, is computed using the velocity fields at the preceding time steps [26]:

$$U_i^* = U_i(t_k) + \left( \frac{3}{2} F_i(t_k) - \frac{1}{2} F_i(t_{k-1}) \right) \Delta t, \tag{16}$$

where the flux, $F_i$, is evaluated as,

$$F_i = -\frac{\partial(U_i U_j)}{\partial x_j} + \frac{1}{\text{Re}} \frac{\partial^2 U_i}{\partial x_j \partial x_j} + \sum_{n=1}^{N_b} S_i^n. \tag{17}$$

Further, the new pressure, $P(t_{k+1})$, is computed by solving its respective Poisson equation in the form,

$$\frac{\partial^2 P(t_{k+1})}{\partial x_j \partial x_j} = \frac{1}{\Delta t} \frac{\partial U_j^*}{\partial x_j}. \tag{18}$$

Equation (18) is solved by iterations by performing, at each iteration step ($j$) the FFT in the horizontal directions and Gaussian elimination in the vertical direction. The iteration procedure stops when the condition $|P_{j+1} - P_j| / P_j < 0.1\%$ is satisfied. Usually this condition is met after $j = 3$–5 iterations. The new velocity at $k + 1$ time step satisfying the incompressibility condition (2) is then computed as:

$$U_i(t_{k+1}) = U_i^* - \frac{\partial P(t_k)}{\partial x_j} \Delta t. \tag{19}$$

At the upper boundary, $\widetilde{\eta} = \eta = 0$, the no-slip (Dirichlet) condition for the velocity is prescribed:

$$\begin{aligned} U_x(\xi, y, 0) &= U_d - cka \, \cos k(x(\xi, 0) - ct), \\ U_y(\xi, y, 0) &= 0, \\ U_z(\xi, y, 0) &= -cka \, \sin k(x(\xi, 0) - ct), \end{aligned} \tag{20}$$

where the surface drift velocity, $U_d$, is expressed as [27]:

$$U_d = c + cka \, \cos k(x(\xi, 0) - ct) - \left[ (c + cka \, \cos k(x(\xi, 0) - ct))^2 - q(2c - q) \right]^{1/2}. \tag{21}$$

Two different cases are considered. In one case, parameter $q$ is put to zero, $U_d = 0$, and the wind stress effects are not taken into account. In this case, the water velocity at the boundary coincides with orbital velocities of the fluid particles in the surface wave. In another case, $q = 0.05c$ is prescribed, and thus $U_d$ is finite so that the wind-stress effects upon the water surface are accounted for.

Periodic conditions for all fields are prescribed at the side boundaries, and the no-slip (Dirichlet) condition for the water velocity is prescribed at the bottom boundary ($\widetilde{\eta} = -1$).

The bubble equation of motion is solved by employing the Adams-Bashforth method:

$$V_i^n(t_{k+1}) = V_i^n(t_k) + \left( \frac{3}{2} F_i^n(t_k) - \frac{1}{2} F_i^n(t_{k-1}) \right) \Delta t \tag{22}$$

$$F_i^n = 3 \frac{DU_i^n}{Dt} + \frac{f(\text{Re}_b^n)}{\tau_n} (U_i^n - V_i^n) + \frac{1}{2} \varepsilon_{ijk} \left( U_j^n - V_j^n \right) \omega_k^n + 2g\delta_{iz} \tag{23}$$

In Equation (23), the surrounding fluid velocity, its acceleration, and the ambient-flow vorticity at the location of each bubble are obtained by a Hermitian 4th-order accuracy

interpolation procedure. The dimensionless bubble response time, $\tau_n$, (or the Stokes number) is expressed as:

$$\tau_n = \frac{d_n^2 U_0}{36 \nu L_0}. \tag{24}$$

The bubble coordinate equation is advanced in time by employing the Adams method as:

$$r_i^n(t_{k+1}) = r_i^n(t_k) + 0.5(V_i^n(t_{k+1}) + V_i^n(t_k))\Delta t \tag{25}$$

The water velocity is initialized as a random, zero-mean field with an amplitude of 0.1% (normalized by the velocity scale, $U_0$). After initiation, a transient occurs during which the velocity field adjusts to boundary conditions, and a statistically stationary flow state is reached (at dimensionless time $t = 100$). The results (not shown) indicate that the the mean, wave velocity field is established comparatively early (~O(10) wave periods) and the transient to the stationary flow state is mainly related to the development of a near-surface, wave-induced turbulent layer (of ~O(100) duration, cf., e.g., [17]). After the stationary flow state is reached, the bubbles are injected into the flow at random locations with a concentration (number density) distribution exponentially decreasing with depth with an e-folding scale close to the surface wave length (similar to void-fraction distributions observed in natural, oceanic conditions [12]). Since bubbles rise due to buoyancy, they reach the upper boundary (i.e., the water surface) and thus leave the computational domain. In order to maintain a constant void fraction throughout the simulation, these bubbles are re-injected at random locations with a spatial distribution exponentially decaying with depth and a velocity equal to the surrounding water velocity.

The simulation of the bubble-laden flow continues until the flow again reaches a stationary state (at $t = 250$) where its statistical characteristics are evaluated. Similar to the previous DNS studies of flows over waved surfaces [19–23,27,28]), in the statistical post-processing analysis, phase averaging, equivalent to averaging over an ensemble of turbulent fluctuations, is performed. This averaging (denoted below by angular brackets) is firstly performed over the $y$-coordinate and time $t$, and further window-averaged over the $\xi$—coordinate over six wave lengths as:

$$\langle F \rangle(\xi, \eta) = \frac{1}{6N_t N_y} \sum_{j=1}^{N_y} \sum_{k=1}^{N_t} \sum_{m=0}^{5} F(\xi + m\lambda, y_j, \eta, t_k), \tag{26}$$

where $F$ is the averaged field, $N_y = 240$, and $N_t = 50$. Time averaging is preformed over interval $250 < t_k \leq 300$, so that the phase of the surface wave at consecutive steps, $t_k$, $t_{k+1}$, changes by $2\pi$ and thus the shape of the upper boundary remains the same for all considered time moments, cf. Equation (10). Further the mean vertical profile is obtained by additional averaging of <$F$> along the $\xi$-coordinate as:

$$[F](\eta) = \frac{1}{N_x/6} \sum_{l=1}^{N_x/6} \langle F \rangle(\xi_l, \eta), \tag{27}$$

where $N_x = 360$. Phase-averaged fields of the bubbles concentration (void fraction) and its fluxes are also evaluated as:

$$\langle C \rangle = \frac{\pi}{6} \sum_{n=1}^{N_d} \frac{d_n^3 w(r^n, r)}{\Omega_g}, \tag{28}$$

$$\langle C V_i^n \rangle = \frac{\pi}{6} \sum_{n=1}^{N_d} \frac{d_n^3 V_i^n w(r^n, r)}{\Omega_g}, \tag{29}$$

and the respective vertical mean profiles, $[C V_i^n]$, are obtained as in Equation (27).

Below the term DNS (i.e., Direct Numerical Simulation) is used, although the surface wave dynamics is prescribed and assumed to be stationary, and the bubbles are consid-

ered as non-deformable and spherical. However, these assumptions are justified for the considered case of non-breaking waves (sufficiently small wave-slope *ka*) and bubble sizes (*d* less than 1 mm). On the other hand, the primitive, full 3D Navier-Stokes equations, Equations (1), (4) and (5), are integrated without employing any closure assumptions.

## 4. Results and Discussion

DNS were performed for the surface wave length $\lambda$ = 15 cm and wave slope *ka* = 0.1 and 0.2 (amplitude $a \approx 0.25$ cm and 0.5 cm). For the chosen length, the wave celerity computed from the linear dispersion relation for surface gravity waves [1] equals $c \approx 49$ cm/s, and the Reynolds number, Equation (2), Re $\approx 73019$. Both wind-driven case (with $q = 0.05c$ and finite surface drift velocity, $U_d$, Equation (21)) and zero surface-stress case, $U_d = 0$, were considered. Bubble diameter, *d*, was varied from 200 to 400 microns, the total number of bubbles in each simulation was maintained constant (up to $N_d = 10^6$ for the smallest, *d* = 200 μm, bubble-size cases) corresponding to a mean (reference) void fraction (concentration, $C_0$) of about $5 \times 10^{-5}$.

### 4.1. Carrier Flow Modification

Figure 2 compares distributions of a vorticity modulus field obtained using DNS in different planes at dimensionless time moment *t* = 300 in the cases with and without bubbles (Figure 2a–c and 2d–f, respectively). The figure shows that in both cases, vorticity streaks, oriented in the direction of the surface wave propagation, are present in the near-surface water layer indicative of turbulence generated by surface wave motion [17]. The figure also shows that rising bubbles create vertical streaks of vorticity in their wakes.

Figure 3 presents distributions of phase-averaged water velocity components, $\langle U_x \rangle$ and $\langle U_z \rangle$, obtained using DNS with wave slope *ka* = 0.1 and 0.2 (left and right panels), both with and without bubbles. The components are compared against an analytical, potential-flow, solution for the velocity field in the surface deep-water gravity wave (in dashed line) [1]. The figure shows that the computed fields agree well with the analytical solution except a near surface layer where the wave-induced turbulence modifies the water velocity. The influence of bubbles on the phase-averaged surface-wave flow field is found to be insignificant (in Figure 3, the full black and blue lines practically coincide).

Mean profiles of the fluctuations of *x,y,z* water-velocity components, $[U_i']$, evaluated as a root mean square deviation,

$$[U_i'] = \left( \left[ U_i^2 \right] - [U_i]^2 \right)^{1/2}, \tag{30}$$

and obtained employing DNS for different bubble diameter, *d* = 200, 300 and 400 μm, with the same reference void fraction ($C_0 \approx 5 \times 10^{-5}$), both with and without wind-induced surface drift, are compared in Figure 4 with the corresponding bubble-free flow cases. The comparison shows that bubbles enhance wave-induced turbulence, and, in both cases (with and without surface wind drift), mostly *z*-component of water velocity is affected (Figure 4c,f). This turbulence enhancement is caused by the presence of wakes induced by rising bubbles (Figure 2f). Figure 4f also indicates that the influence of a wind- driven surface stress synchronizes the mean velocity fluctuations for different bubble sizes. The explanation of this observation requires further research and is to be reported elsewhere.

### 4.2. Bubbles Dispersion

The trajectories of individual bubbles of different sizes (with diameters *d* = 200 μm and 400 μm) obtained using DNS without and with wind-induced surface drift, are presented in Figures 5 and 6, respectively. The figures also show the depth-dependence of the forces imposed on the bubbles by the surrounding water (drag, fluid acceleration, and lift) along the trajectories.

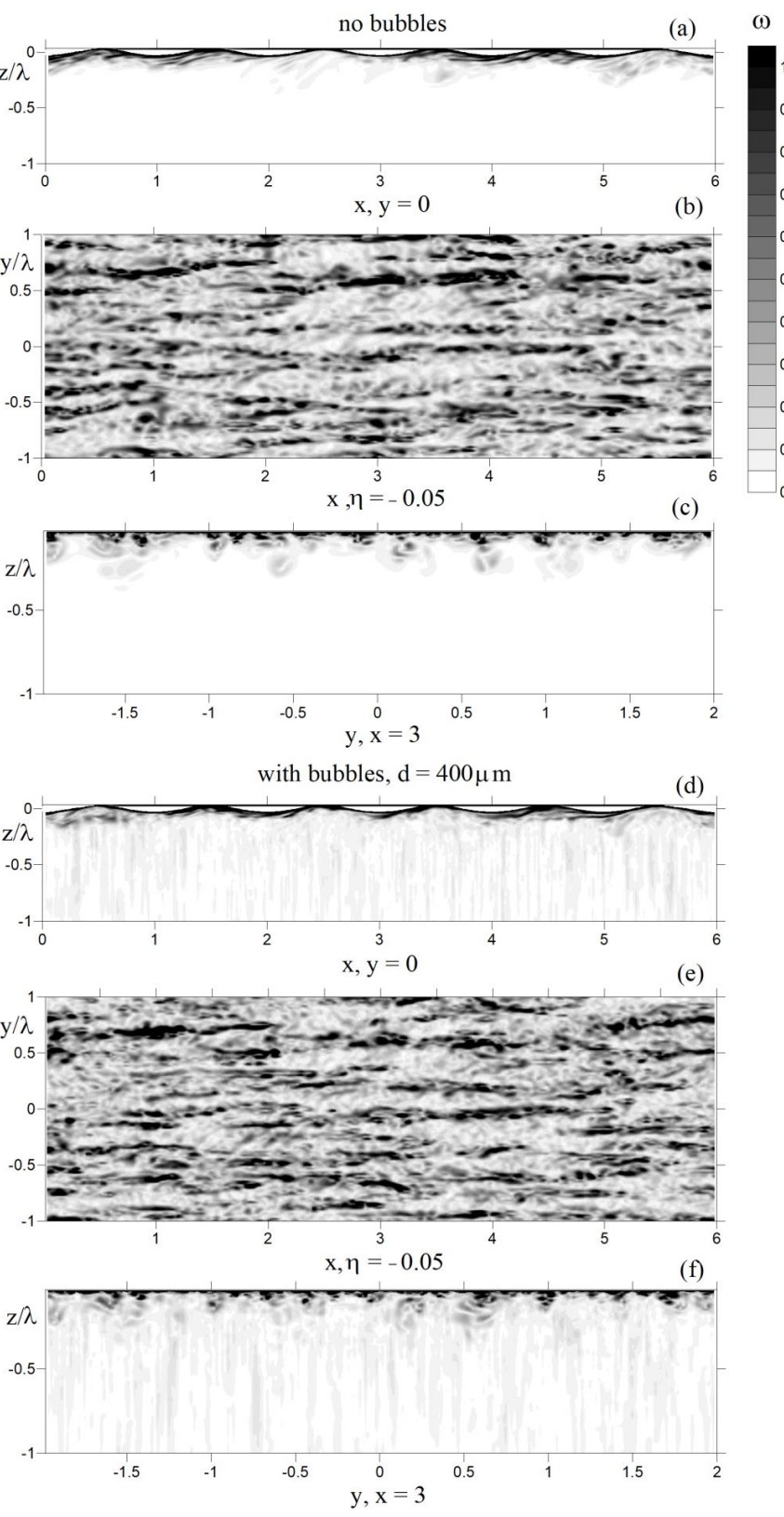

**Figure 2.** Instantaneous distribution of the vorticity modus, $\omega$, obtained using DNS in central $(x,z)$ (**a**,**d**) and $(y,z)$ (**c**,**f**) planes and in a near-surface $(x, y)$ plane at $\eta = -0.05$ (**b**,**e**). Panels (**a**–**c**) are for the no-bubbles case, and panels (**d**–**f**) are for DNS with bubbles with diameter $d = 400$ μm. In both cases, surface drift velocity, Equation (21), is added at the waved boundary accounting for the wind stress. Wave slope $ka = 0.2$.

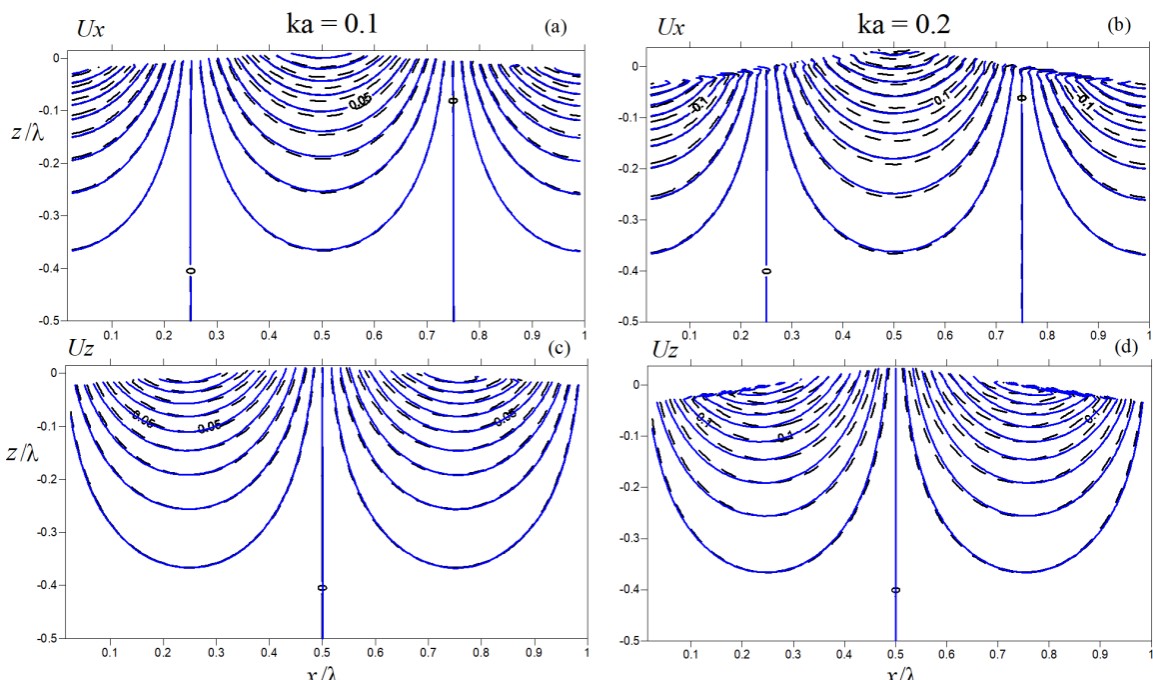

**Figure 3.** Phase-averaged water velocity components, $U_x$ (**a**,**b**) and $U_z$ (**c**,**d**), obtained using DNS with wave slope $ka = 0.1$ (**a**,**c**) and $ka = 0.2$ (**b**,**d**) with and without bubbles (black and blue lines, respectively). Dashed line shows analytical solution for the deep-water surface wave [1]; $U_d = 0$.

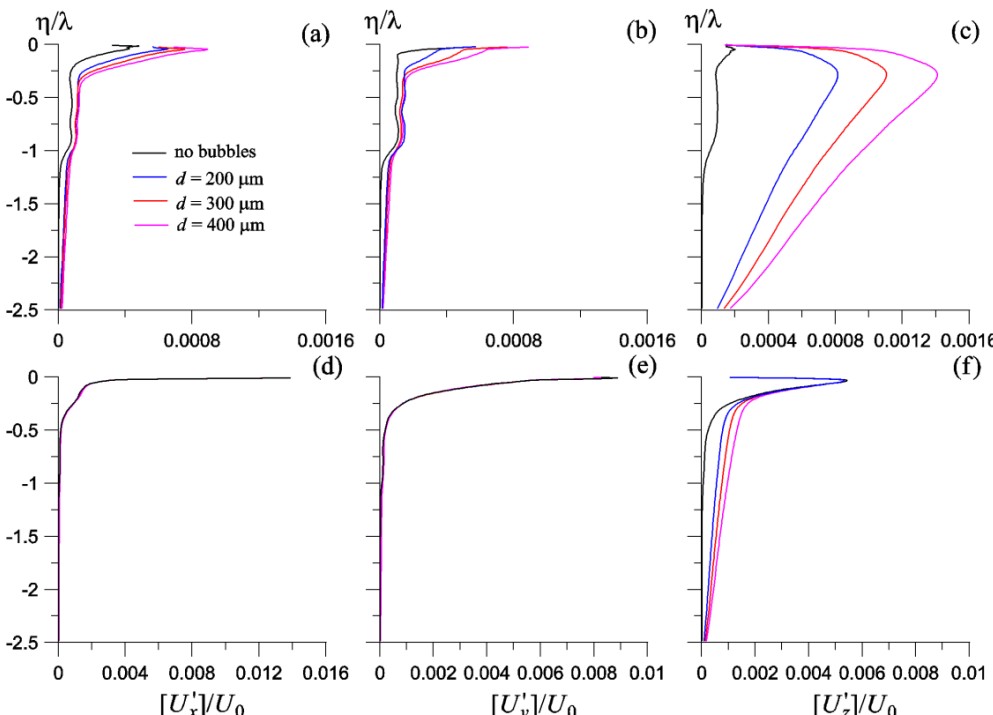

**Figure 4.** Mean profiles of water velocity fluctuations obtained using DNS with wave slope $ka = 0.2$ and different bubbles diameter, $d$. Panels (**a**–**c**) are for the case $U_d = 0$ and panels (**d**–**f**) are for the non-zero surface drift case.

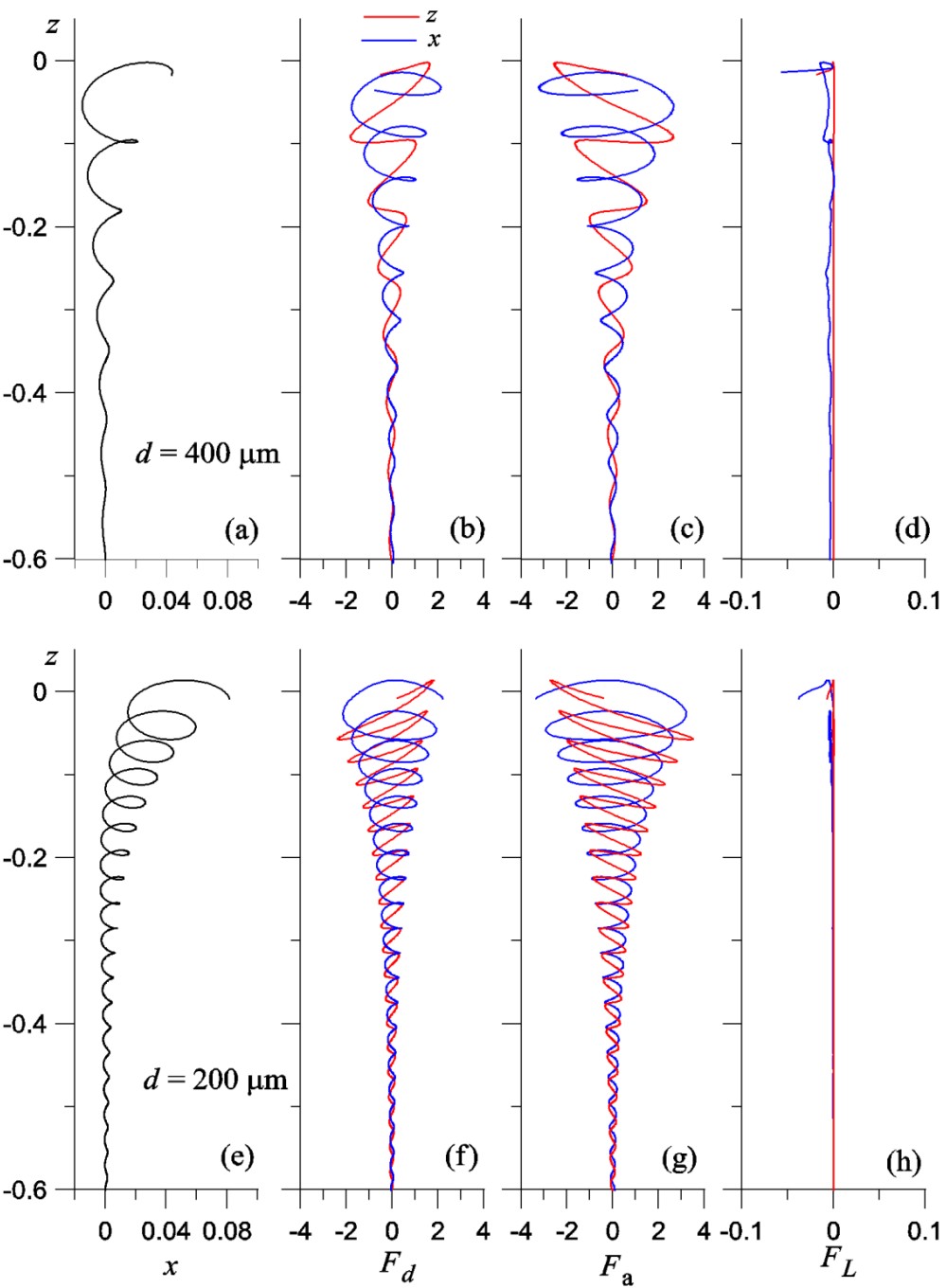

**Figure 5.** Trajectories of individual bubbles (**a,e**) and $x$ and $z$ components of forces imposed on the bubble by the surrounding water [drag $F_d$ (**b,f**); fluid acceleration $F_a$ (**c,g**); lift $F_L$ (**d,h**)] obtained using DNS without surface-induced wind drift, $U_d = 0$. Top and bottom panels are for the bubble diameter $d = 400$ and $200$ μm, respectively. Wave slope $ka = 0.2$. For convenience, in panels (**b,f**) the mean value of the $z$-component of $F_d$ is subtracted.

Figures 5 and 6 show that the influence of the surface-wave-induced motions on the bubbles dynamics increases as they rise toward the water surface: the trajectories, being almost straight-vertical at $z/\lambda < 0.5$, oscillate with an increasing amplitude as the depth decreases and start spiraling sufficiently close to the surface (Figures 5a,e and 6a,e). The bubbles experience a mean drift in the direction of the surface wave propagation. This drift is similar to the classical Stokes drift of the Lagrangian fluid particles in a surface wave and caused by the decreasing dependence of the wave motion amplitude on depth [29]. As

expected, the drift becomes more pronounced in the presence of the wind-induced surface drift (Figure 6). In the considered case, however, this drift is also modified by the bubbles' ascent due to buoyancy, so that larger bubbles ($d$ = 400 μm) are less subject to this mean Lagrangian drift since their terminal rise velocity is almost 4 times larger as compared to 200 μm- bubbles (cf. Figure 7 below). As a result, their dynamics are less affected by the oscillatory wave-field motion. In this aspect, there is some analogy between the observed reduction of the drift due to bubble rising and the "crossing trajectories" effect governing the dispersion of inertial particles by turbulence, where particles' settling reduces their dispersion by ambient turbulence [30].

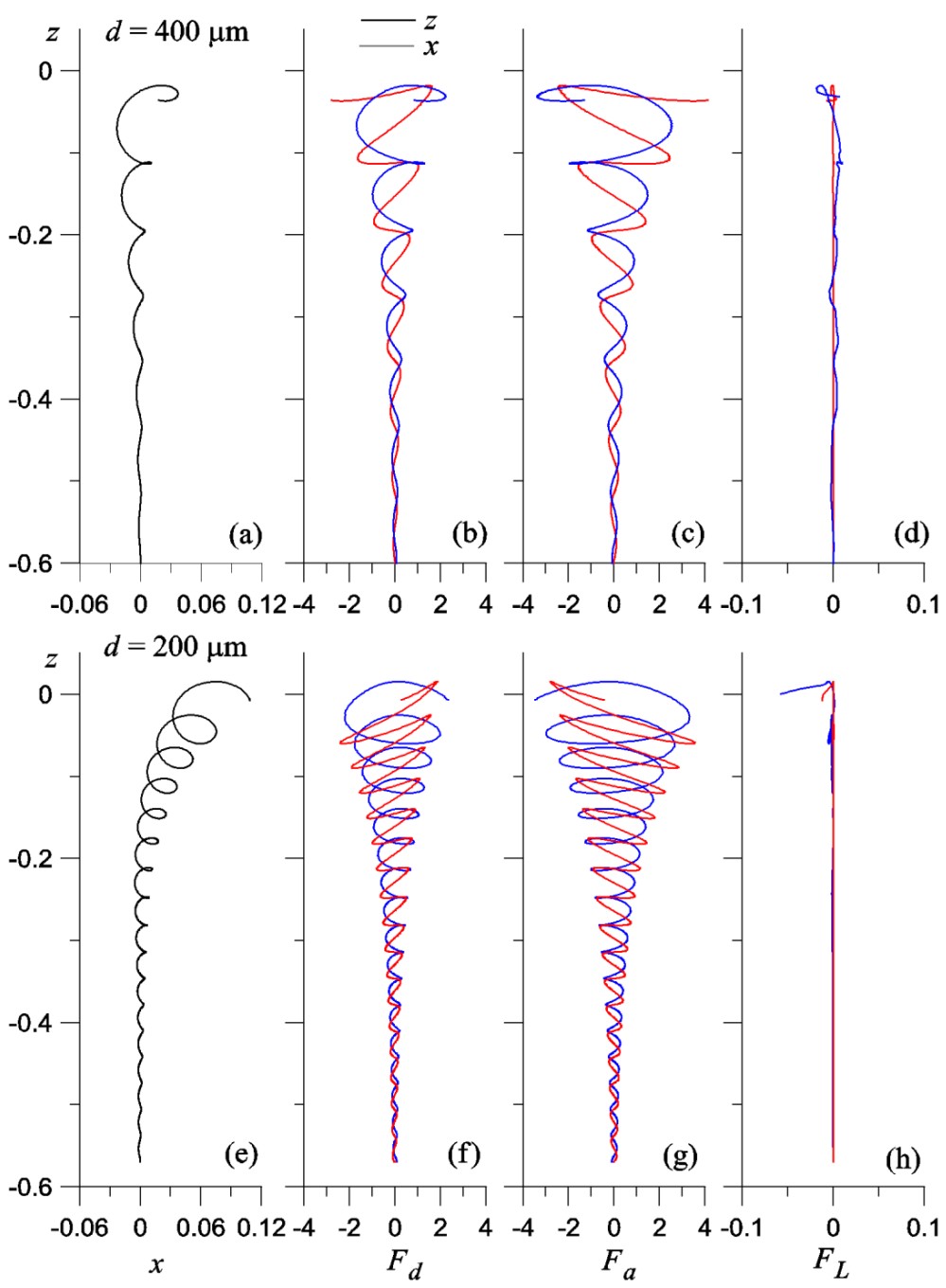

**Figure 6.** The same as in Figure 5 but obtained using DNS with finite $U_d$.

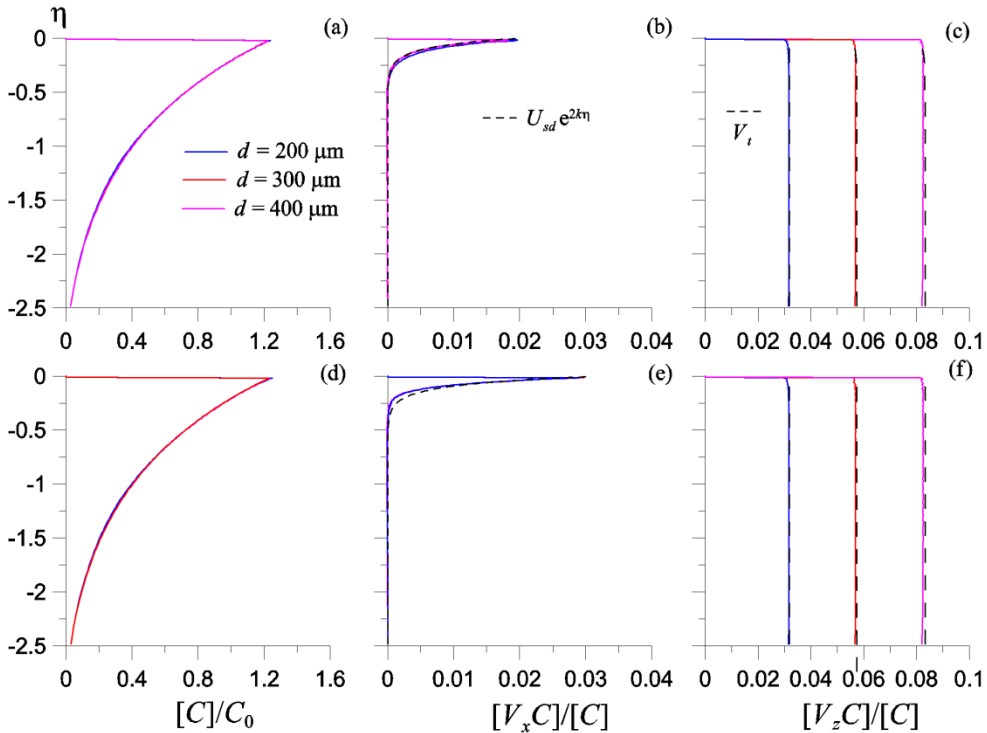

**Figure 7.** Profiles of mean bubble concentration, $[C]$, (**a,d**); and horizontal (**b,e**) and vertical (**c,f**) components of the concentration (void fraction) fluxes, $[CV_x]$ and $[CV_z]$, normalized with $[C]$, obtained in DNS for different bubble size, $d$, (200 μm, blue color; 300 μm, red; and 400 μm, magenta). Top panels (**a–c**) are for the cases without the wind-induced surface drift ($U_d = 0$), and bottom panels (**d–f**) are for finite $U_d$. Parameterizations for the fluxes in panels (**b,c**) and (**e,f**), Equations (31) and (32), are shown in dotted and dashed line, respectively. Wave slope $ka = 0.2$.

The analysis of forces governing bubbles dynamics reveals that the drag force, $F_d$, and the fluid-acceleration force, $F_a$, are both of the same order, and much larger as compared to the lift force, $F_L$. The forces, $F_a$ and $F_d$, increase monotonically as the bubbles rise toward the water surface, whereas the lift force becomes non-zero only in close vicinity of the surface, where the fluid motion is not strictly irrotational due to the presence of wave-induced turbulence [17]. The profiles of the mean bubble concentration (or void fraction), $[C]$, and $x$ and $z$ components of the concentration fluxes, $[CV_x]$ and $[CV_z]$, were evaluated in DNS according to Equations (28) and (29) for different bubble size, and both with and without wind stress at the water surface (Figure 7). Figure 7 also compares numerical solution for the fluxes with their parameterizations in the form:

$$[CV_x] = V_{sd}[C]\exp(-2kz), \tag{31}$$

$$[CV_z] = V_t[C], \tag{32}$$

where $V_{sd}$ is the bubble drift velocity at the water surface ($V_{sd} \approx 0.02$ and $V_{sd} \approx 0.03$ for the zero (Figure 7b) and non-zero (Figure 7e) wind-induced-drift cases, respectively); $V_t$ is a terminal velocity determined by numerical solution of the following equation:

$$0 = -\frac{f(\mathrm{Re}_b)}{\tau}V_t + 2g\delta_{iz}, \tag{33}$$

which is obtained from the bubble equation of motion, Equation (5), rewritten for a bubble rizing with constant (terminal) velocity, $V_t$, in a quiescent water. Equation is solved by the Newton's method [26].

Figure 7 shows that in our numerical experiment, the bubble concentration decreases exponentially with depth, as observed in natural oceanic conditions [12]. Thus, the injection mechanism employed in our DNS adequately reproduces the required void-fraction vertical distribution.

The parameterization for the horizontal concentration flux, $[CV_x]$, in the absence of the wind-induced surface drift ($U_d = 0$, (b), Equation (31)) is obtained using the Stokes-drift solution [1] adapted here for a somewhat smaller surface drift velocity ($V_{sd} \approx 0.02$) as compared to the classical surface Stokes-drift velocity ($c(ka)^2 \approx 0.04$ for the considered surface-wave slope $ka = 0.2$). Note that a similar effect of a reduced Stokes drift velocity was observed in DNS of a free-propagating, non-breaking surface waves [31]. In the presence of the wind-induced surface drift ($U_d \approx 0.04$), the same parameterization is used but with $V_{sd} \approx 0.03$. Both parameterizations agree well with DNS results.

The vertical component of the void-fraction flux, $[CV_z]$, is well predicted by Equation (32) in both (zero and non-zero wind induced drift) cases. That means that neither wave-induced irrotational motions nor turbulence affect on average the bubble rising rate, so that the void fraction vertical flux is analogous to that in a quiescent water.

## 5. Conclusions

We have presented an algorithm for numerical modeling of microbubble dispersion in the near-surface water layer of the upper ocean, under the action of non-breaking, progressive surface waves. The algorithm is based on a Eulerian-Lagrangian approach where full, 3D Navier-Stokes equations for the carrier flow induced by a waved water surface are solved in a Eulerian frame, and the trajectories of individual bubbles are simultaneously tracked in a Lagrangian frame, taking into account the impact of the bubbles on the carrier flow and the impact of a wind-induced surface drift. The bubbles diameters are considered in the range from 200 to 400 microns (thus, micro-bubbles), and the effects related to the bubbles deformation and dissolution in water are neglected. The wave shape is prescribed and assumed to be unaffected by either bubbles or induced turbulent motions.

The simulations results show that bubbles are capable of enhancing the carrier-flow turbulence, as compared to the bubble-free flow, and that the vertical water velocity fluctuations are mostly augmented, and increasingly so by larger bubbles. The results also show that the bubbles dynamics are governed by buoyancy, the surrounding fluid acceleration force, and the drag force whereas the impact of the lift force on the bubble dynamics remains negligible. On the basis of the simulation results, parameterizations for the void fraction fluxes have been obtained. The results show that the vertical component of the void-fraction flux remains unaffected by either the wave motion or wave-induced turbulence as compared to that in a quiescent water. The horizontal void-fraction flux is produced by a mean drift of the bubbles in the direction of the surface wave propagation and can be regarded as analogous to the Stokes drift of Lagrangian (non-inertial) particles in a 2D surface wave modified by the bubbles' ascent. The developed algorithm and parameterizations for void-fraction fluxes can be used for prediction of microbubble dispersion in the ocean upper layer and further employed in large-scale prognostic models.

As a next step in the development of the algorithm, a more complicated problem is to be considered where the wave motion is fully resolved (i.e., not prescribed) thus allowing modification of the surface wave by either induced turbulence or bubbles. This however remains a subject for future research.

**Author Contributions:** Algorithm development, performing DNS, data analysis, writing—original draft preparation, O.A.D.; methodology, formal analysis, writing—review and editing, W.-T.T. All authors have read and agreed to the published version of the manuscript.

**Funding:** This work is supported by the Russian-Taiwanese Joint Research Project through the Russia RFBR Grant (No. 21-55-52005) and the Taiwan MOST Grant (MOST 110-2923-M-002-014 –MY3). OD

is additionally supported by RFBR Grant (No. 20-05-00322,) and by the Ministry of Education and Science of the Russian Federation (Task No. 0030-2019-0020).

**Institutional Review Board Statement:** Not applicable.

**Informed Consent Statement:** Not applicable.

**Data Availability Statement:** Not applicable.

**Conflicts of Interest:** The authors declare no conflict of interest.

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
