# Peer review of "Numerical Simulation of Micro-Bubbles Dispersion by Surface Waves"

_algorithms, doi:10.3390/a15040110_

Round 1

Reviewer 1 Report

Review report for article no. Algorithms-1634607.

Figure 2 is illegible without much enlargement. You can place one photo below the other and enlarge both of them.

DNS is a method, so writing "in DNS" is incomplete in my opinion. Instead, e.g. using DNS. This note applies to several places in the paper.

Couldn't it be possible to add colors in figure 5 and enlarge the entire curtain by 50%? This gray color is not visible. The same is true for figure 6.

These authors cited quite a lot of contemporary literature. A deeper literature review would be a plus.

Author Response

Please, see the attached reply.

Reviewer 2 Report

The paper seems interesting but it fails to address why is necessary to have such a study, especially that the authors are addressing sea water surface. Why shall we see microbubbles at sea ?

The introduction is totally irrelevant to support the scope of the paper and thus I am not sure why is the work needed?

I believe that the introduction improvement will make the paper acceptable for publication.

Author Response

Please, see the attached reply.

Reviewer 3 Report

Please see reviewer's comments for authors uploaded as a pdf file

Author Response

Please, see the attached reply. 

Round 2

Reviewer 1 Report

All my comments have been taken into account. In my opinion, this paper looks better compared to the previous version. I recommend this paper for publication. 

Author Response

We are grateful to the reviewer for the positive review.

Reviewer 2 Report

Although the introduction was slightly improved I still cannot understand all the efforts on doing this paper.

there is no clear practical message for the paper.

furthermore, the conclusion are talking about carrier-flow turbulence which is not shown in the paper.

first and last paragraph of the conclusions are opposite. Have the authors considered the wave motion or not?

Author Response

Please, see the attached reply.

Reviewer 3 Report

The authors have addressed my reviewer comments well.  Thank you.

Author Response

We sincerely thank the reviewer for the positive report.

Round 3

Reviewer 2 Report

All my comments have been responded.